# REASONING WITH SAMPLING:
# YOUR BASE MODEL IS SMARTER THAN YOU THINK

**Aayush Karan**
Harvard University
akaran1@g.harvard.edu

**Yilun Du**
Harvard University
ydu@seas.harvard.edu

## ABSTRACT

Frontier reasoning models have exhibited incredible capabilities across a wide array of disciplines, driven by posttraining large language models (LLMs) with reinforcement learning (RL). However, despite the widespread success of this paradigm, much of the literature has been devoted to disentangling truly novel behaviors that emerge during RL but are not present in the base models. In our work, we approach this question from a different angle, instead asking whether comparable reasoning capabilities can be elicited from base models at inference time by *pure sampling*, *without any additional training*. Inspired by Markov chain Monte Carlo (MCMC) techniques for sampling from sharpened distributions, we propose a simple iterative sampling algorithm leveraging the base models' own likelihoods. Over different base models, we show that our algorithm offers substantial boosts in reasoning that nearly match and even outperform those from RL on a wide variety of single-shot tasks, including MATH500, HumanEval, and GPQA. Moreover, our sampler avoids the collapse in diversity over multiple samples that is characteristic of RL-posttraining. Crucially, our method does not require training, curated datasets, or a verifier, suggesting broad applicability beyond easily verifiable domains.

## 1 INTRODUCTION

Reinforcement learning (RL) has become the dominant paradigm for enhancing the reasoning capabilities of large language models (LLMs) (Guo et al., 2025; Hu et al., 2025). Equipped with a reward signal that is typically automatically verifiable, popular RL techniques have been successfully applied to posttrain frontier models, leading to sizeable performance gains in domains like math, coding, and science (Hendrycks et al., 2021; Li et al., 2022; Rein et al., 2024).

Despite the widespread empirical success of RL for LLMs, a large body of literature has centered around the following question: are the capabilities that emerge during RL-posttraining *fundamentally novel behaviors* that are *not* present in the base models? This is the question of *distribution sharpening* (He et al., 2025; Shao et al., 2025; Yue et al., 2025): that is, whether the posttrained distribution is simply a "sharper" version of the base model distribution, instead of placing mass on reasoning traces the base model is unlikely to generate.

Several works point towards the difficulty in learning new capabilities with RL-posttraining. He et al. (2025); Song et al. (2025) compare the pass@$k$ (multi-shot) scores of base models with posttrained models, finding that for large $k$, base models actually outperform while the latter suffer from degraded generation diversity. In such cases, RL appears to redistribute pass@$k$ performance to single-shot performance at the expense of multi-shot reasoning. Yue et al. (2025) also notes that the reasoning traces post-RL are tightly concentrated at high likelihoods/confidences under the base model, seemingly drawing from existing high-likelihood capabilities. We illustrate this point in our own experiments in Figure 4. Regardless, the advantage of RL-posttraining for single-shot reasoning has remained, as of yet, undeniable.

In this paper, we present a surprising result: *sampling directly from the base model can achieve single-shot reasoning capabilities on par with those from RL.*

We propose a sampling algorithm for base models that leverages additional compute at inference time, achieving single-shot performance that *nearly matches* RL-posttraining on *in-domain* reason-

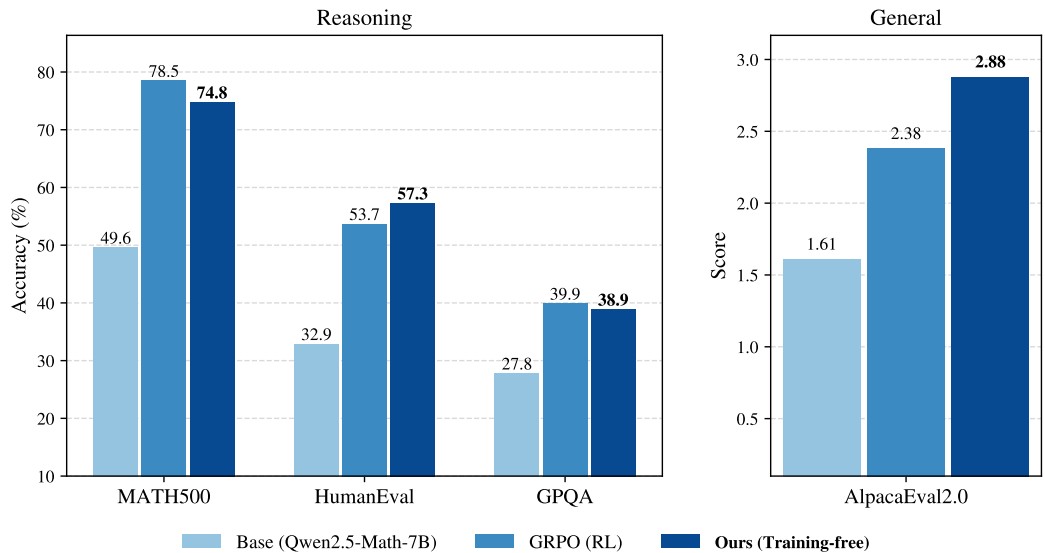

Figure 1: **Our sampling algorithm can match and outperform RL-posttraining.** Left: we compare our sampling algorithm (ours) against the base model (base) and RL-posttraining (GRPO) on three *verifiable reasoning* tasks (MATH500, HumanEval, GPQA). Right: we compare them on an *unverifiable general task* (AlpacaEval2.0). Our algorithm achieves comparable performance to GRPO within the posttraining domain (MATH500) but can *outperform* on out-of-domain tasks such as HumanEval and AlpacaEval.

ing tasks and can even *outperform* on *out-of-domain* reasoning tasks. Furthermore, we observe that generation diversity *does not degrade* with our sampler; in fact, our pass@$k$ (multi-shot) performance *strongly outperforms* RL. We benchmark specifically against Group Relative Policy Optimization (GRPO), which is the standard RL algorithm for enhancing LLM reasoning (Shao et al., 2024).

Crucially, our algorithm is *training-free*, *dataset-free*, and *verifier-free*, avoiding some of the inherent weaknesses of RL methods including extensive hyperparameter sweeps to avoid training instabilities, the need to curate a diverse and expansive posttraining dataset, and the lack of guaranteed access to a ground truth verifier/reward signal (Prabhudesai et al., 2025).

Our contributions can be summarized as follows:

 i) We introduce the *power distribution* as a useful sampling target for reasoning tasks. Since it can be explicitly specified with a base LLM, no additional training is required.

 ii) We further introduce an approximate sampling algorithm for the power distribution using a Markov chain Monte Carlo (MCMC) algorithm that iteratively resamples token subsequences according to their base model likelihoods.

iii) We empirically demonstrate the effectiveness of our algorithm over a range of models (Qwen2.5-Math-7B, Qwen2.5-7B, Phi-3.5-mini-instruct) and reasoning tasks (MATH500, HumanEval, GPQA, AlpacaEval 2.0). Our results show that sampling directly from the base model can achieve results on par with GRPO. In fact, for some out-of-domain tasks, our algorithm consistently *outperforms* the RL baseline. Moreover, over multiple samples, we avoid the collapse in diversity afflicting RL-posttraining, achieving the best of both worlds in terms of single-to-few-shot reasoning capabilities as well as sample diversity.

Our results collectively illustrate that existing base models are much more capable at single-shot reasoning than current sampling methods reveal.

## 2    RELATED WORKS

**Reinforcement learning for LLMs.** RL has been instrumental in posttraining LLMs. Early on, RL with human feedback (RLHF) (Ouyang et al., 2022) was developed as a technique to align LLMs with human preferences using a trained reward model. Recently, RL with verifiable rewards (RLVR)

has emerged as a powerful new posttraining technique, where many works (Guo et al., 2025; Lambert et al., 2024; Hu et al., 2025; Zeng et al., 2025) discovered that a simple, end-of-generation reward given by an automated verifier could substantially enhance performance on difficult reasoning tasks in mathematics and coding. The Group Relative Policy Optimization (GRPO) algorithm was at the center of these advances (Shao et al., 2024). Building off of this success, many subsequent works have examined using reward signals derived from internal signals such as self-entropy (Zhao et al., 2025), confidence (Prabhudesai et al., 2025), and even random rewards (Shao et al., 2025). Similar to these works, our paper examines base model likelihoods as a mechanism for improving reasoning performance, but crucially, our technique is *training-free*.

**Autoregressive MCMC sampling with LLMs.** Prior works have explored integrating classic MCMC techniques with autoregressive sampling. Many settings including red-teaming, prompt-engineering, and personalized generation can be framed as targeting sampling from the base LLM distribution but *tilted* towards an external reward function. Zhao et al. (2024) proposes learning intermediate value functions that are used in a *Sequential Monte Carlo* (SMC) framework (Chopin, 2004), where multiple candidate sequences are maintained and updated according to their expected future reward. Similarly, Faria et al. (2024) proposes a *Metropolis-Hastings* (MH) algorithm, which instead of maintaining multiple candidates performs iterative resampling, again updating according to expected reward. Methodologically, our sampling algorithm is most similar to this latter work, but the crucial difference is that our target sampling distribution is completely specified by the base LLM, *avoiding the need for an external reward.*

## 3    PRELIMINARIES

Let $\mathcal{X}$ be a finite vocabulary of tokens, and let $\mathcal{X}^T$ denote the set of finite sequences of tokens $x_{0:T} = (x_0, x_1, \ldots, x_T)$, where $x_i \in \mathcal{X}$ for all $i$ and $T \in \mathbb{Z}_{\geq 0}$ is some nonnegative integer. For convenience, for a given $t$, let $x_{<t} = (x_0, \ldots, x_{t-1})$ and $x_{>t} = (x_{t+1}, \ldots, x_T)$, with similar definitions for $x_{\leq t}$ and $x_{\geq t}$. In general, $\mathbf{x}$ refers to a token sequence $x_{0:T}$, where $T$ is implicitly given.

Then an LLM defines a distribution $p$ over token sequences $\mathcal{X}^T$ by autoregressively learning the conditional token distributions $p(x_t|x_{<t})$ for all $t$, giving the *joint distribution* via the identity

$$p(x_{0:T}) = \prod_{t=0}^{T} p(x_t|x_{<t}). \tag{1}$$

To sample a sequence from $p$, we simply sample from the LLM token by token using the conditional distributions, which by (1) directly samples from the joint distribution.

## 4    MCMC SAMPLING FOR POWER DISTRIBUTIONS

In this section, we introduce our sampling algorithm for base models. Our core intuition is derived from the notion of distribution sharpening posed in Section 1. *Sharpening* a reference distribution refers to reweighting the distribution so that high likelihood regions are further upweighted while low likelihood regions are downweighted, biasing samples heavily towards higher likelihoods under the reference. Then if RL posttrained models really are just sharpened versions of the base model, we should be able to explicitly specify a target sampling distribution that achieves the same effect.

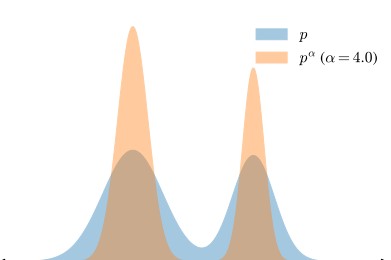

Figure 2: **A toy example of distribution sharpening.** Here $p$ is a mixture of Gaussians, which we plot against $p^\alpha$ ($\alpha = 4.0$).

We organize this section as follows. Section 4.1 presents this target sharpened distribution and provides some mathematical motivation for why its samples are amenable for reasoning tasks. Section 4.2 introduces a general class of Markov chain Monte Carlo (MCMC) algorithms aimed at actually sampling from this target distribution, and finally, Section 4.3 details our specific implementation for LLMs.

### 4.1 REASONING WITH POWER DISTRIBUTIONS

One natural way to sharpen a distribution $p$ is to sample from the *power distribution* $p^\alpha$. Since

$$p(\mathbf{x}) > p(\mathbf{x}') \implies \frac{p(\mathbf{x})^\alpha}{p(\mathbf{x}')^\alpha} > \frac{p(\mathbf{x})}{p(\mathbf{x}')} \qquad (\alpha \in [1, \infty]), \tag{2}$$

it follows that exponentiating $p$ *increases* the relative weight on higher likelihood sequences ($\mathbf{x}$) while *decreasing* the relative weight on lower likelihood ones ($\mathbf{x}'$) (see Figure 2 for a visualization).

A related but well-known sharpening strategy is *low-temperature sampling* (Wang et al., 2020), which exponentiates the conditional next-token distributions at each step:

$$p_{\text{temp}}(x_t | x_0 \ldots x_{t-1}) = \frac{p(x_t | x_{t-1} \ldots x_0)^\alpha}{\sum_{x_t' \in \mathcal{X}} p(x_t' | x_{t-1} \ldots x_0)^\alpha}, \tag{3}$$

where the *temperature* is $\tau = 1/\alpha$. A common misconception is that sampling with (3) over $T$ tokens is equivalent to sampling from $p^\alpha$; however, this is false in a subtle yet crucial way, as we illuminate in the following.

**Proposition 1.** *Low-temperature sampling does not sample from the power distribution $p^\alpha$.*

*Proof.* We show that the associated conditional next-token distributions are distinct at each timestep $t$. The conditional distribution on $x_t$ for $p^\alpha$ is given by

$$p_{\text{pow}}(x_t | x_0 \ldots x_{t-1}) = \frac{\sum_{x_{>t}} p(x_0, \ldots, x_t, \ldots, x_T)^\alpha}{\sum_{x_{\geq t}} p(x_0, \ldots, x_t, \ldots, x_T)^\alpha}. \tag{4}$$

Using Bayes rule

$$p(x_t | x_{t-1} \ldots x_0) = \frac{p(x_0, \ldots, x_t)}{p(x_0, \ldots, x_{t-1})} = \frac{\sum_{x_{>t}} p(x_0, \ldots, x_t, \ldots, x_T)}{\sum_{x_{\geq t}} p(x_0, \ldots, x_t, \ldots, x_T)}, \tag{5}$$

we can rewrite the low-temperature marginal (3) as

$$p_{\text{temp}}(x_t | x_0 \ldots x_{t-1}) = \frac{\left(\sum_{x_{>t}} p(x_0, \ldots, x_t, \ldots, x_T)\right)^\alpha}{\sum_{x_t'} \left(\sum_{x_{>t}} p(x_0, \ldots, x_t, \ldots, x_T)\right)^\alpha}. \tag{6}$$

Ignoring normalizations for clarity, the relative weight on token $x_t$ for sampling from $p^\alpha$ is given by a *sum of exponents*

$$p_{\text{pow}}(x_t | x_{<t}) \propto \sum_{x_{>t}} p(x_0, \ldots, x_t, \ldots, x_T)^\alpha. \tag{7}$$

Meanwhile, the relative weight for low-temperature sampling is given by an *exponent of sums*

$$p_{\text{temp}}(x_t | x_{<t}) \propto \left(\sum_{x_{>t}} p(x_0, \ldots, x_t, \ldots, x_T)\right)^\alpha. \tag{8}$$

Since the relative weights of next-token prediction are distinct for each sampling strategy, it follows that the joint distribution over seqeunces must also be distinct for each sampler. Hence, the distribution on sequences given by low-temperature sampling is *not the same* as the one given by $p^\alpha$. $\square$

One intuitive way to understand this difference is that low-temperature sampling does not account for how exponentiation sharpens the likelihoods of "future paths" at time step $t$, instead "greedily" averaging all these future likelihoods (*exponent of sums* (8)). On the other hand, sampling from $p^\alpha$ *inherently accounts* for future completions as it exponentiates all future paths (*sum of exponents* (7)) before computing the weights for next-token prediction. This has the following consequence:

**Observation 1.** *The power distribution upweights tokens with few but high likelihood future paths, while low-temperature sampling upweights tokens with several but low likelihood completions.*

**Example 1.** We can observe this phenomenon with a simple example. Let us consider the token vocabulary $\mathcal{X} = \{a, b\}$ and restrict our attention to two-token sequences $(x_0, x_1)$: $aa, ab, ba, bb$. Let

$$p(aa) = 0.00, \qquad p(ab) = 0.40, \qquad p(ba) = 0.25, \qquad p(bb) = 0.25,$$

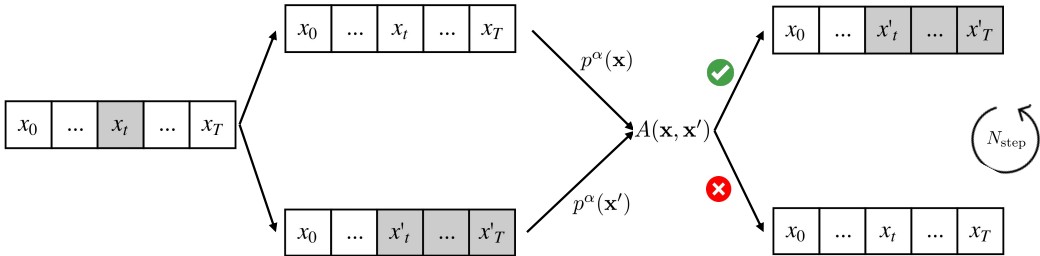

Figure 3: **Illustrating Metropolis-Hastings with random resampling.** A random index $t$ is selected and a new candidate is generated by resampling. Based on the relative likelihoods, the candidate is accepted or rejected, and the process repeats.

so that

$$p(x_0 = a) = 0.40, \qquad p(x_0 = b) = 0.50.$$

Let $\alpha = 2.0$. Under $p^\alpha$, we have

$$p_{\text{pow}}(x_0 = a) \propto 0.00^2 + 0.40^2 = 0.160, \qquad p_{\text{pow}}(x_0 = b) \propto 0.25^2 + 0.25^2 = 0.125,$$

so $p^\alpha$ prefers sampling $a$ over $b$. Under low-temperature sampling,

$$p_{\text{temp}}(x_0 = a) \propto (0.00 + 0.40)^2 = 0.160, \qquad p_{\text{temp}}(x_0 = b) \propto (0.25 + 0.25)^2 = 0.250,$$

preferring sampling $b$ over $a$. If $p^\alpha$ samples $x_0 = a$, there is only one future path with likelihood $0.40$. If $p_{\text{temp}}$ samples $x_0 = b$, there are two future paths $ba, bb$, but either choice has likelihood $0.25$.

In other words, even though $a$ has lower conditional likelihood under both $p$ and $p_{\text{temp}}$, $p^\alpha$ upweights $a$ and samples the highest likelihood two-token sequence. $b$ has many future paths contributing to a higher likelihood under $p$ and $p_{\text{temp}}$, but leads to low likelihood sequences. We provide a stronger formalization of this phenomenon in Appendix A.2.

Thus, sampling from $p^\alpha$ encourages sampling tokens which have fewer but higher likelihood "future paths", as opposed to tokens with several lower likelihood completions. This type of behavior is immensely valuable for reasoning tasks. For example, choosing "wrong" tokens that have high average likelihoods but trap outputs in low likelihood individual futures are examples of *critical windows* or *pivotal tokens* (Li et al., 2025; Abdin et al., 2024), a phenomenon where a few tokens are highly influential in the correctness of language model outputs. In fact, sharp critical windows have been shown to correlate strongly with reasoning failures (Li et al., 2025). Instead, embedded in sampling from the power distribution is an implicit bias towards planning for future high likelihood tokens.

## 4.2 THE METROPOLIS-HASTINGS ALGORITHM

Now that we have seen how sampling from $p^\alpha$ can in theory assist the underlying LLM's ability to reason, our aim now turns towards proposing an algorithm to accurately sample from it. Given an LLM $p$, we have access to the values $p^\alpha$ over any sequence length; however, these values are *unnormalized*. Direct sampling from the true probabilities requires normalizing over all sequences $(x_0, \ldots, x_T) \in \mathcal{X}^T$, which is computationally intractable.

To get around this, we invoke a Markov Chain Monte Carlo (MCMC) algorithm known as Metropolis-Hastings (MH) (Metropolis et al., 1953), which targets exactly what we want: approximate sampling from an unnormalized probability distribution. The MH algorithm constructs a Markov chain of sample sequences $(\mathbf{x}^0, \mathbf{x}^1, \ldots, \mathbf{x}^n)$ using an arbitrary *proposal distribution* $q(\mathbf{x}|\mathbf{x}^i)$ to select the next candidate $\mathbf{x}^{i+1}$. With probability

$$A(\mathbf{x}, \mathbf{x}^i) = \min \left\{ 1, \frac{p^\alpha(\mathbf{x}) \cdot q(\mathbf{x}^i|\mathbf{x})}{p^\alpha(\mathbf{x}^i) \cdot q(\mathbf{x}|\mathbf{x}^i)} \right\}, \tag{9}$$

candidate $\mathbf{x}$ is accepted as $\mathbf{x}^{i+1}$; otherwise, MH sets $\mathbf{x}^{i+1} = \mathbf{x}^i$. This algorithm is especially convenient as it only requires the relative weights given by $p^\alpha$ (as the normalization weights in $A$ cancel) and works with *any* generic but tractable sampler $q$ with minimal restrictions. Remarkably, for large enough $n$, this process converges to sampling from the *target distribution* $p^\alpha$ under the following (quite minimal) conditions on the proposal distribution (Neal, 1993):

**Definition 1.** The proposal distribution $q$ is *irreducible* if for any set $X$ with nonzero mass under the target distribution $p^\alpha$, $q$ has nonzero probability of eventually sampling from $X$. The proposal is *aperiodic* if the induced chain of samples does not return to the same sample after a fixed interval number of steps.

Thus, we must simply ensure that our proposal distribution satisfies irreducibility and aperiodicity, and Metropolis-Hastings takes care of the rest. On a practical level, we would also like both $q(\mathbf{x}|\mathbf{x}^i)$ and its reverse $q(\mathbf{x}^i|\mathbf{x})$ to be easily computable.

Consider the following family of *random resampling* proposal distributions (see Figure 3). Let $p_{\text{prop}}$ be a proposal LLM. With uniform probability $\frac{1}{T}$, select a random $t \in [1, T]$ and *resample the sequence* starting at index $t$ using $p_{\text{prop}}$. Then the transition likelihood $q(\mathbf{x}|\mathbf{x}^i)$ is simply the likelihood of the resampling. Note that at each candidate selection step, we have a nonzero probability of transitioning between any two sequences $\mathbf{x}, \mathbf{x}' \in \mathcal{X}$, since with some probability we can always resample as early as the beginning of $\mathbf{x}$. This ensures our proposal distribution is both irreducible and aperiodic. Moreover, $q(\mathbf{x}^i|\mathbf{x})$ is easy to calculate by symmetry, since we can treat $\mathbf{x}^i$ as a resampled version of $\mathbf{x}$.

With the flexibility endowed by Metropolis-Hastings, we can choose the proposal LLM $p_{\text{prop}}$ to be any LLM with any sampling strategy (e.g., low-temperature sampling).

### 4.3 POWER SAMPLING WITH AUTOREGRESSIVE MCMC

A direct implementation of Metropolis-Hastings for LLMs would involve initializing with a sampled token sequence of length $T$, subsequently generating new candidates of length $T$ with (9) over many, many iterations. This process is computationally expensive, however, due to the repeated, full sequence inference calls to the LLM.

In fact, the main downside to MCMC algorithms in practice is the potential for an *exponential mixing time* (Gheissari et al., 2017), where a poor choice of initialization or proposal distribution can result in an exponentially large number of samples required before convergence to the target distribution. This problem is exacerbated if the sample space has *high dimensionality* (Bandeira et al., 2022; Schmidler & Woodard, 2013), which is precisely exhibited by the sequence space of tokens $\mathcal{X}^T$, especially for long sequences/large values of $T$.

To remedy this, we propose an algorithm that leverages the sequential structure of autoregressive sampling. We define a series of intermediate distributions which we progressively sample from, until converging to the target distribution $p^\alpha$. In particular, samples from one intermediate distribution initiate a Metropolis-Hastings process for the next, helping avoid pathological initializations.

Fix block size $B$ and proposal LLM $p_{\text{prop}}$, and consider the sequence of (unnormalized) distributions

$$\emptyset \longrightarrow p(x_0, \dots, x_B)^\alpha \longrightarrow p(x_0, \dots, x_{2B})^\alpha \longrightarrow \cdots \longrightarrow p(x_0, \dots, x_T)^\alpha, \tag{10}$$

where $p(x_0, \dots, x_{kB})$ denotes the joint distribution over token sequences of length $kB$, for any $k$. For convenience, let $\pi_k$ denote the distribution given by

$$\pi_k(x_{0:kB}) \propto p(x_{0:kB})^\alpha. \tag{11}$$

Suppose we have a sample from $\pi_k$. To obtain a sample from $\pi_{k+1}$, we initialize a Metropolis-Hastings process by sampling the next $B$ tokens $x_{kB+1:(k+1)B}$ with $p_{\text{prop}}$. We subsequently run the MCMC sampling procedure for $N_{\text{MCMC}}$ steps, using the *random resampling* proposal distribution $q$ from the previous section. The full details are presented in Algorithm 1.

Note that Algorithm 1 is *single-shot*: even though multiple inference calls are made, the decision to accept vs. reject new tokens is made purely by base model likelihoods to simulate sampling *a single sequence* from $p^\alpha$. We can interpret this as a new axis for *inference-time scaling*, as we expend additional compute during sampling to obtain a higher quality/likelihood sample.

To quantify the scaling, we can estimate the average number of tokens generated by Algorithm 1. Note that each candidate generation step when sampling from $\pi_k(x_{0:kB}$ resamples an average of $\frac{kB}{2}$ tokens, $N_{\text{MCMC}}$ times. Summing over all $k$, the expected number of tokens generated is

$$\mathbb{E}_{\text{tokens}} = N_{\text{MCMC}} \sum_{k=1}^{\lceil T/B \rceil} \frac{kB}{2} \approx \frac{N_{\text{MCMC}} T^2}{4B}. \tag{12}$$

---

**Algorithm 1:** Power Sampling for Autoregressive Models

---

**Input**       : base $p$; proposal $p_{\text{prop}}$; power $\alpha$; length $T$

**Hyperparams:** block size $B$; MCMC steps $N_{\text{MCMC}}$

**Output**     : $(x_0, \ldots, x_T) \sim p^\alpha$

1  **Notation:** Define the unnormalized intermediate target

$$\pi_k(x_{0:kB}) \;\propto\; p(x_{0:kB})^\alpha.$$

2  **for** $k \leftarrow 0$ **to** $\lceil \frac{T}{B} \rceil - 1$ **do**

3     |  Given prefix $x_{0:kB}$, we wish to sample from $\pi_{k+1}$. Construct initialization $\mathbf{x}^0$ by extending autoregressively with $p_{\text{prop}}$:

$$x_t^{(0)} \sim p_{\text{prop}}\big(x_t \mid x_{<t}\big), \qquad \text{for } kB + 1 \leq t \leq (k+1)B.$$

    |  Set the current state $\mathbf{x} \leftarrow \mathbf{x}^0$.

4     |  **for** $n \leftarrow 1$ **to** $N_{\text{MCMC}}$ **do**

5     |    |  Sample an index $m \in \{1, \ldots, (k+1)B\}$ uniformly.

6     |    |  Construct proposal sequence $\mathbf{x}'$ with prefix $x_{0:m-1}$ and resampled completion:

$$x_t' \sim p_{\text{prop}}\big(x_t \mid x_{<t}\big), \qquad \text{for } m \leq t \leq (k+1)B.$$

7     |    |  Compute acceptance ratio (9)

$$A(\mathbf{x}', \mathbf{x}) \;\leftarrow\; \min \left\{ 1, \; \frac{\pi_k(\mathbf{x}')}{\pi_k(\mathbf{x})} \cdot \frac{p_{\text{prop}}(\mathbf{x} \mid \mathbf{x}')}{p_{\text{prop}}(\mathbf{x}' \mid \mathbf{x})} \right\}.$$

    |    |  Draw $u \sim \text{Uniform}(0, 1)$;

8     |    |  **if** $u \leq A(\mathbf{x}', \mathbf{x})$ **then accept** and set $\mathbf{x} \leftarrow \mathbf{x}'$

9     |  **end**

10    |  Set $x_{0:(k+1)B} \leftarrow \mathbf{x}$ to fix the new prefix sequence for the next stage.

11  **end**

12  **return** $x_{0:T}$

---

The key tradeoff here is between the block size $B$ and number of MCMC steps $N_{\text{MCMC}}$. A larger $B$ requires larger "jumps" between intermediate distributions, requiring a larger $N_{\text{MCMC}}$ to adequately transition. In Section 5, we empirically find a value for $B$ that makes Algorithm 1 performant for relatively small values of $N_{\text{MCMC}}$.

## 5  EXPERIMENTS

### 5.1  EXPERIMENTAL SETUP

**Evaluation.** We use a standard suite of reasoning benchmarks ranging across mathematics, coding, and STEM (MATH500, HumanEval, GPQA), along with a non-verifiable benchmark (AlpacaEval 2.0) evaluating general helpfulness. We evaluate all of our methods and baselines *single-shot*; i.e., on one final response string.

- **MATH500**: The MATH dataset (Lightman et al., 2024) consists of competition math problems spanning seven categories including geometry, number theory, and precalculus. There are 12500 problems total, with 7500 training problems and 5000 test problems. MATH500 is a specific randomly chosen subset of the test set standardized by OpenAI.
- **HumanEval**: HumanEval is a set of 164 handwritten programming problems covering algorithms, reasoning, mathematics, and language comprehension (Chen et al., 2021). Each problem has an average of 7.7 associated unit tests, where solving the problem corresponds to passing all unit tests.
- **GPQA**: GPQA (Rein et al., 2024) is a dataset of multiple-choice science questions (physics, chemistry, and biology) which require advanced reasoning skills to solve. We use subset GPQA Diamond for evaluation, which consists of 198 questions which represent the highest quality subset of the GPQA dataset.
- **AlpacaEval 2.0**: The AlpacaEval dataset is a collection of 805 prompts (Dubois et al., 2024) that gauge general helpfulness with questions asking e.g., for movie reviews, recommendations,

|  | MATH500 | HumanEval | GPQA | AlpacaEval2.0 |
|---|---|---|---|---|
| **Qwen2.5-Math-7B** | | | | |
| Base | 0.496 | 0.329 | 0.278 | 1.61 |
| Low-temperature | 0.690 | 0.512 | 0.353 | 2.09 |
| **Power Sampling (ours)** | **0.748** | **0.573** | **0.389** | **2.88** |
| GRPO (MATH) | **0.785** | **0.537** | **0.399** | **2.38** |
| **Qwen2.5-7B** | | | | |
| Base | 0.498 | 0.329 | 0.278 | 7.05 |
| Low-temperature | 0.628 | 0.524 | 0.303 | 5.29 |
| **Power Sampling (ours)** | **0.706** | **0.622** | **0.318** | **8.59** |
| GRPO (MATH) | **0.740** | **0.561** | **0.354** | **7.62** |
| **Phi-3.5-mini-instruct** | | | | |
| Base | 0.400 | 0.213 | 0.273 | 14.82 |
| Low-temperature | 0.478 | 0.585 | 0.293 | **18.15** |
| **Power Sampling (ours)** | **0.508** | **0.732** | **0.364** | **17.65** |
| GRPO (MATH) | **0.406** | **0.134** | **0.359** | **16.74** |

Table 1: **Power sampling (ours) matches and even outperforms GRPO across model families and tasks.** We benchmark the performance of our sampling algorithm on MATH500, HumanEval, GPQA, and AlpacaEval 2.0. We bold the scores of both our method and GRPO, and underline whenever our method outperforms GRPO. Across models, we see that power sampling is comparable to GRPO on in-domain reasoning (MATH500), and can outperform GRPO on out-of-domain tasks.

> and reading emails. The model responses are graded by an automated LLM judge (GPT-4-turbo), which determines a preference for the model responses over those from a baseline (also GPT-4-turbo). The resulting score is a win rate of model responses normalized for the length of the model response.

**Models.** To demonstrate the efficacy of our sampling algorithm, we use the base models Qwen2.5-Math-7B, Qwen2.5-7B, and Phi-3.5-mini-instruct. For our RL baselines, we use the implementation of GRPO in Shao et al. (2025), which posttrains these models on the MATH training split. For both the Qwen2.5 models, we use the default hyperparameters used to benchmark their performance in Shao et al. (2025). For the Phi-3.5 model, we use a set of hyperparameters selected from Abdin et al. (2024) that avoids training instabilities and converges to improvement over the base model over a large number of epochs.

**Sampling Algorithm.** For our implementation of power sampling (Algorithm 1), we set the maximum $T$ to be $T_{\max} = 3072$ (termination can happen earlier with an EOS token) and block size $B = 3072/16 = 192$. Empirically, we find $\alpha = 4.0$ coupled with a proposal LLM $p_{\mathrm{prop}}$ chosen as the base model with sampling temperature $1/\alpha$ to be most performant for reasoning tasks. For AlpacaEval 2.0, we find that having a proposal distribution of higher temperature ($\tau = 0.5$) improves performance.

## 5.2 RESULTS

**Main results.** We display our main results in Table 1. Across base models of different families, our sampling algorithm achieves massive, near-universal boosts in single-shot accuracies and scores over different reasoning and evaluation tasks that reach, e.g., up to **+51.9%** on HumanEval with Phi-3.5-mini and **+25.2%** on MATH500 with Qwen2.5-Math. In particular, on MATH500, which is in-domain for RL-posttraining, power sampling achieves accuracies that are on par with those obtained by GRPO. Furthermore, on out-of-domain reasoning, our algorithm again matches GRPO on GPQA and actually *outperforms* on HumanEval by up to **+59.8%**. Similarly, power sampling consistently outperforms on the non-verifiable AlpacaEval 2.0, suggesting a *generalizability* of our boosts to domains beyond verifiability.

The surprising success of this fundamentally simple yet training-free sampling algorithm underscores the latent reasoning capabilities of existing base models.

*Filter an input list of strings only for ones that start with a given prefix.* (Phi-3.5-mini-instruct: HumanEval)

| Method | Response | Passed |
|---|---|---|
| Ours | ```return [s for s in strings`
`          if s.startswith(prefix)]``` | true |
| GRPO | ```return [string for string in strings if`
`          string.startswith(f'{prefix}'*2)]``` | false |

Table 2: **Sample responses on HumanEval: Phi-3.5-mini-instruct.** We present an example where our method solves a simple coding question, but GRPO does not.

### 5.3 ANALYSIS

We analyze how the reasoning characteristics of power sampling relate to those of GRPO. We present an example in Table 2, with further examples in Appendix A.5.

**Reasoning trace likelihoods and confidences.** By design, power sampling targets sampling higher likelihood sequences from the base model. In Figure 4, the left graph plots a histogram of the output sequence log-likelihoods (averaged by length) of the base model, power sampling, and GRPO responses on MATH500, where likelihoods are taken relative to the Qwen2.5-Math-7B base model. Our method samples from higher likelihood regions of the base model, as intended, but still maintains noticeable spread. Meanwhile, GRPO samples are heavily concentrated at the highest likelihood peak.

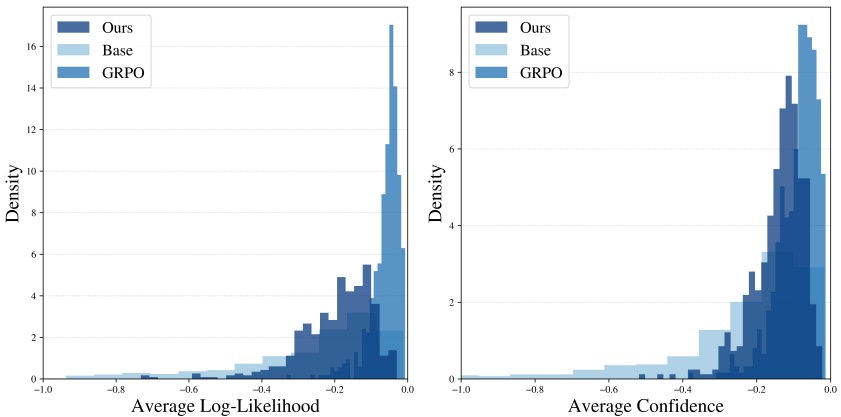

Figure 4: **Base model (Qwen2.5-Math-7B) likelihoods and confidences for MATH500 responses.** Left: We plot the log-likelihoods (relative to the base model) of original, power sampling, and GRPO responses over MATH500. Right: We do the same but for confidences relative to the base model. We observe that GRPO samples from the highest likelihood and confidence regions with power sampling close behind, which correlates with higher empirical accuracy.

We also plot the base model *confidence* of MATH500 responses, defined to be the average negative entropy (*uncertainty*) of the next-token distributions (Prabhudesai et al., 2025):

$$\text{Conf}(x_{0:T}) = \frac{1}{T+1}\sum_{t=0}^{T}\sum_{x \in \mathcal{X}} p(x|x_{<t}) \log p(x|x_{<t}). \tag{13}$$

The right plot of Figure 4 demonstrates that our method's and GRPO responses sample from similarly high confidence regions from the base model, which again correspond to regions of higher likelihood and correct reasoning.

**Reasoning trace lengths.** Another defining characteristic of RL-posttraining is long-form reasoning (Guo et al., 2025), where samples tend to exhibit longer responses. On MATH500, Qwen2.5-Math-7B averages a response length of **600** tokens, while GRPO averages **671** tokens. Surprisingly, power sampling achieves a similar average length of **679** tokens, *without explicitly being encouraged to favor longer generations.* This emerges naturally from the sampling procedure.

**Diversity and pass@$k$ performance.** Again, notice the peaked and highly concentrated likelihoods/confidences of GRPO relative to the distributional spread of power sampling in Figure 4. This suggests GRPO exhibits a collapse in diversity while our sampler does not, aligning with the obser-

vation that RL-posttraining strongly sharpens the base model distribution at the expense of diversity (Song et al., 2025). To quantify the comparative diversity of power sampling relative to GRPO, we can plot the pass@$k$ accuracy rate, where a question is solved if at least one of $k$ samples is accurate. Figure 5 shows exactly this: unlike GRPO, whose pass@$k$ performance tapers off for large $k$, power sampling strongly outperforms for $k > 1$. Moreover, our performance curve supersedes that of the base model until finally converging in performance. In particular, we are able to achieve GRPO-level single-shot performance *without compromising multi-shot performance* (see Appendix A.4 for other domains), addressing a long-standing downside to RL-posttraining.

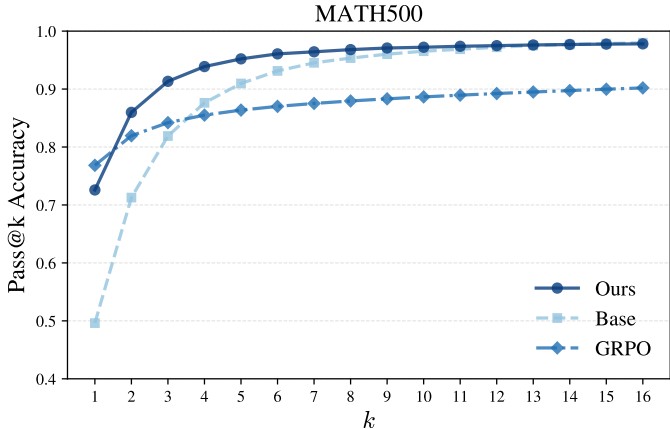

Figure 5: **Pass@$k$ performance on MATH500**. We plot the pass@$k$ accuracy (correct if at least one of $k$ samples is accurate) of power sampling (ours) and RL (GRPO) relative to the base model (Qwen2.5-Math-7B). Our performance curve is strictly better than both GRPO and the base model, and our pass rate at high $k$ matches the base model, demonstrating sustained generation diversity.

## 6 CONCLUSION

In this work, we present an algorithm that samples directly from a base model without any additional training or access to an external signal, achieving a single-shot reasoning performance that is on par with, and sometimes even better than, that of a state-of-the-art RL-posttraining algorithm. We use the discussion of RL distribution sharpening to motivate defining the power distribution as a valuable target distribution for reasoning. Although exact power distribution sampling is intractable, we employ classic MCMC techniques alongside the sequential structure of autoregressive generation to define our power sampling algorithm, which demonstrates strong empirical performance.

Our results suggest that base model capabilities are underutilized at sampling time and point towards a close relationship between high likelihood regions of the base model and strong reasoning capabilities. Employing additional compute at sampling-time with a stronger understanding of base model capabilities offers a promising direction for expanding the scope of reasoning beyond verifiability.

## 7 ACKNOWLEDGEMENTS

We gratefully acknowledge support from the Paul and Daisy Soros Foundation, the Kempner Insitute, the DoD NDSEG Fellowship, Pickle Robotics, and the Amazon AGI Lab.

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

# A APPENDIX

## A.1 MORE RELATED WORKS

**Annealed sampling for diffusion.** In the statistical physics and Monte Carlo literature, sampling from $p^\alpha$ is known as sampling from an *annealed*, or *tempered*, distribution (Neal, 1998) and has inspired a new wave of interest within the diffusion community. Indeed, in traditional MCMC sampling, annealing is used as a way to avoid mode-collapse during sampling and more accurately sample from complex multimodal distributions (Łatuszyński et al., 2025). This has re-emerged as inference-time sampling methods for diffusion that aim to steer a pretrained model towards "tilted distributions" (Du et al., 2023; Kim et al., 2025; Karan et al., 2025; Wang et al., 2025; Kong et al., 2025; Zhang et al., 2025). Where traditional RL techniques exhibit mode collapse, applications in the physical sciences (Sambridge, 2014) require multimodal sampling. To this end, works such as Du et al. (2023); Wang et al. (2025); Kim et al. (2025) construct sequences of annealed distributions to ease the transition from base diffusion distribution to tilted distribution. Other works (Skreta et al., 2025; Xu et al., 2025) intentionally target sampling from $p^\alpha$ for $\alpha > 1$ as a means of generating higher quality samples from the base diffusion model, which is particularly popular for generating more designable proteins (Geffner et al., 2025).

## A.2 ADDITIONAL THEORETICAL DISCUSSION

In this section, we provide a stronger formalization of the phenomenon that power sampling down-weights tokens that trap outputs in low-likelihood futures while low-temperature sampling does not.

**Proposition 2** (*Informal*). *Power sampling upweights tokens with small support but high likelihood completions, while low-temperature sampling upweights tokens with large support but low likelihood completions.*

**Definition 2.** For the rest of this section, fix a prefix $x_{0:t-1}$. We say that $x_t$ has **marginal weight** $\varepsilon$ under the conditional next-token distribution if $\sum_{x>t} p(x_0, \ldots, x_t, \ldots x_T) = \varepsilon$.

We consider a simplified model of the "critical window" or "pivotal token" phenomenon (Li et al., 2025; Abdin et al., 2024), which refers to intermediate tokens that strongly influence the quality of the final generation. We differentiate between pivotal tokens that lead to high-likelihood futures vs. low-likelihood ones.

**Definition 3.** At one extreme, a pivotal token maximally induces a high-likelihood completion if it places its entire marginal weight $\varepsilon$ on one future (singular support); i.e., for only one choice of $x > t$ is $p(x_0, \ldots, x_t, \ldots, x_T)$ nonzero. We call such a token a **positive pivotal token.**

**Definition 4.** At the other extreme, a pivotal token minimizes the likelihood of any future if its entire marginal weight $\varepsilon$ is uniformly distributed across $N$ future completions. In other words, there exist $N$ completions $x > t$ such that $p(x_0, \ldots, x_t, \ldots, x_T)$ are all nonzero with likelihood $\frac{\varepsilon}{N}$. We call such a token a **negative pivotal token.**

Our simplified model of high and low-likelihood futures examines when positive pivotal tokens are favored over negative pivotal tokens under a given sampling distribution. In particular, we show that power sampling can upweight a positive pivotal token over a negative one even if the latter has a higher marginal weight, whereas low-temperature sampling always upweights the negative pivotal token in such a scenario.

Of course, whenever a positive pivotal token has higher marginal weight, both power sampling and low-temperature sampling will upweight it.

**Proposition 3.** *Let $x_t$ be a positive pivotal token with marginal weight $\varepsilon$, and let $x'_t$ be a negative pivotal token with marginal weight $\varepsilon'$ and support $N$. Then if*

$$\frac{\varepsilon'}{N^{1-1/\alpha}} < \varepsilon < \varepsilon',  \tag{14}$$

*the future likelihood of $x_t$ is higher than any future likelihood of $x'_t$. Moreover, power sampling upweights $x_t$ over $x'_t$ while low-temperature sampling upweights $x'_t$ over $x_t$.*

*Proof.* Since $\alpha \geq 1$, it follows that

$$\frac{\varepsilon'}{N^{1-1/\alpha}} > \frac{\varepsilon'}{N}  \tag{15}$$

and thus $\varepsilon > \frac{\varepsilon'}{N}$, establishing that the future completion likelihood of $x_t$ is greater than that of $x_t'$ (i.e. the assignment of positive and negative pivotal tokens is consistent).

Now, if $\varepsilon < \varepsilon'$, then under the low-temperature distribution, the relative marginal weights on $x_t$ and $x_t'$ are $\varepsilon^\alpha$ and $\varepsilon'^\alpha$, so the probability of choosing $x_t$ is *downweighted* relative to $x_t'$. However, for the power distribution, the relative marginal weights are $p_{\text{pow}}(x_t|x_{<t}) = \varepsilon^\alpha$ and $p_{\text{pow}}(x_t'|x_{<t}) = \frac{\varepsilon'^\alpha}{N^{\alpha-1}}$. Then, as long as $\varepsilon^\alpha > \frac{\varepsilon'^\alpha}{N^{\alpha-1}} \iff \varepsilon > \frac{\varepsilon'}{N^{1-1/\alpha}}$, token $x_t$ will be *upweighted* relative to token $x_t'$.

In other words, the marginal weight on $x_t$ can be *less than* the mass on $x_t'$ under $p$, but if the completion for $x_t$ has higher likelihood than any individual completion for $x_t'$, power sampling favors $x_t$ over $x_t'$. □

## A.3 Hyperparameters for Power Sampling

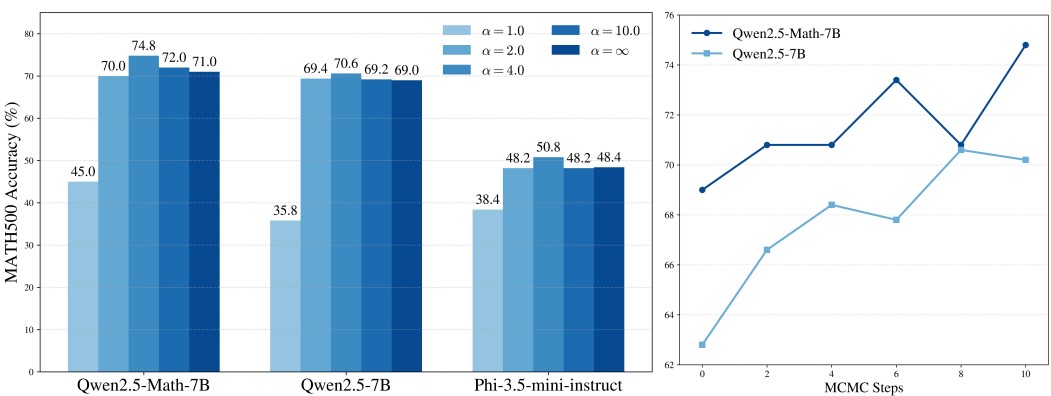

Figure 6: **Effect of hyperparameters on power sampling.** Left: We plot MATH500 accuracy across model families for various values of $\alpha$. Right: We plot the increase in accuracy of power sampling on Qwen models as the number of MCMC steps increases.

**The effect of power distributions.** The two most important hyperparameters for power sampling are the choice of $\alpha$ and the number of MCMC (resampling) steps during sequence generation $N_{\text{MCMC}}$. At the extremes, choosing $\alpha = 1.0$ samples from the base model directly, while taking $\alpha \to \infty$ has the effect of deterministically accepting any resampled sequence that strictly increases the likelihood. Of course, even though higher base model likelihoods correlate with better reasoning (Figure 4), directly optimizing for likelihood is not necessarily optimal for reasoning, suggesting an ideal intermediate value of $\alpha$.

In Figure 6, we display MATH500 accuracies across various values of $\alpha$ and find that an intermediate $\alpha = 4.0$ outperforms other values, as expected. Noticeably, the accuracies of power sampling remain relatively stable beyond $\alpha \geq 2.0$, suggesting that power sampling in practice is relatively robust to the choice of $\alpha$.

**Test-time scaling with MCMC steps.** On the other hand, $N_{\text{MCMC}}$ toggles the inference-time compute expended by our algorithm, providing a natural axis for test-time scaling. In Section 4.3 we raised the notion of a *mixing time*, or the number of MCMC steps required before adequately sampling from the target distribution. In our case, we expect that the fewer MCMC steps we take, the further our algorithm samples from the target $p^\alpha$.

We plot performance dependence on $N_{\text{MCMC}}$ in Figure 6 and notice a steady increase in accuracy until $N_{\text{MCMC}} = 10$, beyond which accuracy remains roughly stable (not plotted). The accuracy difference from using fewer MCMC steps is noticeable but no more than 3-4% between $N_{\text{MCMC}} = 2$ and $N_{\text{MCMC}} = 10$. However, the jump in accuracy by using at least two steps as opposed to none is substantial (3-4%).

We can even compute the total amount of tokens generated by our method relative to running GRPO. From (12), our sampler generates $\frac{1}{4B} \cdot N_{\text{MCMC}}T$ times as many tokens as standard inference to generate a sequence of length $T$. Plugging in our experimental parameters $N_{\text{MCMC}} = 10$, $T = 679$ (our

average output length for MATH500) and $B = 192$, running inference with power sampling incurs a multiplier of **8.84**× the number of tokens as running standard inference. Since GRPO generates multiple rollouts per example during training, *our method incurs roughly the same inference cost as one epoch of GRPO training*, assuming 8 rollouts per sample with identical dataset sizes. Typically though, one GRPO epoch is still more expensive as it uses 16 rollouts and a training set that is larger than MATH500.

### A.4    PASS@K ACCURACIES OVER MULTIPLE DOMAINS

In this section, we plot the pass@$k$ performance of power sampling, GRPO, and the base model (Qwen2.5-Math-7B) over MATH500, GPQA, and HumanEval to demonstrate that our sampling algorithm is highly performant at both single-shot and multi-shot reasoning while maintaining response diversity. Power sampling is plotted with $\alpha = 4.0$ for MATH500 and GPQA and $\alpha = 1.67$ for HumanEval (this temperature exhibits slightly better results at earlier $k$). In all cases, both in-domain and out-of-domain for GRPO, power sampling has near universally better performance than both GRPO and the base model in pass@$k$ for $k > 1$ and matches, if not exceeds, the base model upper bound at large $k$.

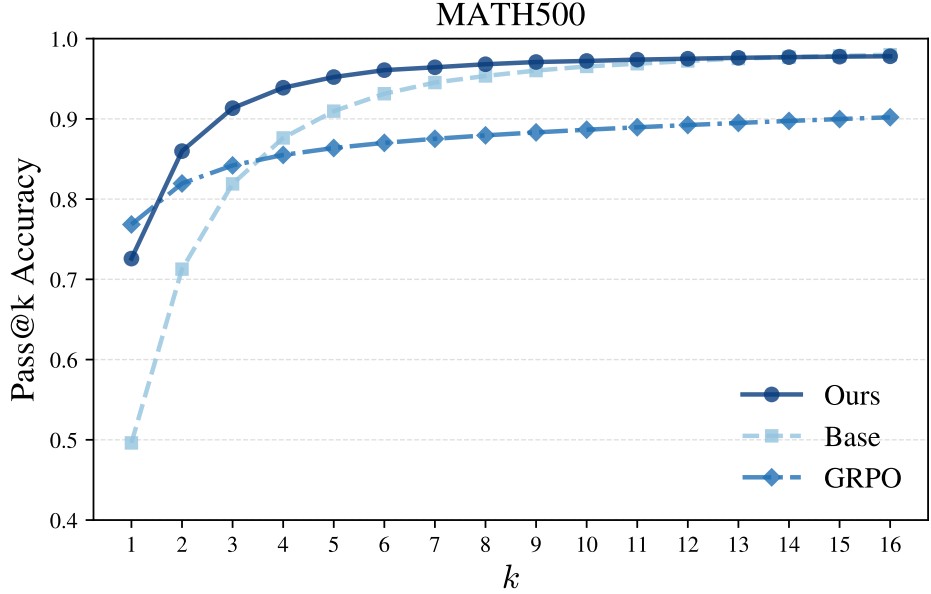

Figure 7: **Pass@$k$ performance on MATH500 (Qwen2.5-Math-7B)**.

One thing to note about these plots is that the loss in diversity varies noticeably from benchmark to benchmark. MATH500 and GPQA clearly show that GRPO has a significantly lower pass@$k$ performance and diversity even for smaller $k$, while on HumanEval GRPO exhibits better pass@$k$ than the base model until $k = 16$. We speculate this might be due to the fact that while MATH500 and GPQA are graded on a "singular" answer, HumanEval is based on completing code where multiple solutions might be accepted, indicating a reduction in diversity may emerge but at much higher $k$. This of course varies model-to-model; in fact, from Table 1, GRPO leads to a reduction in performance on HumanEval relative to the base model Phi 3.5 mini. We noticed that a majority of failures are at least due to formatting issues of the base model being sustained: $76.05\%$ of failures for GRPO are marked for failing to adhere to unindents, compared to $81.40\%$ for the base model. Such errors do not constitute failures of power sampling however.

### A.5    MORE QUALITATIVE EXAMPLES

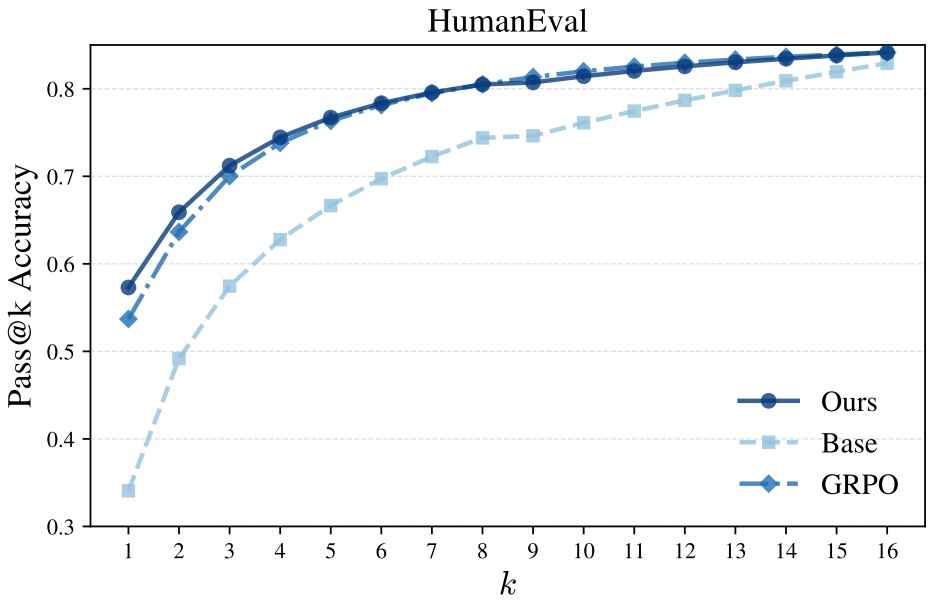

Figure 8: **Pass@$k$ performance on HumanEval (Qwen2.5-Math-7B)**.

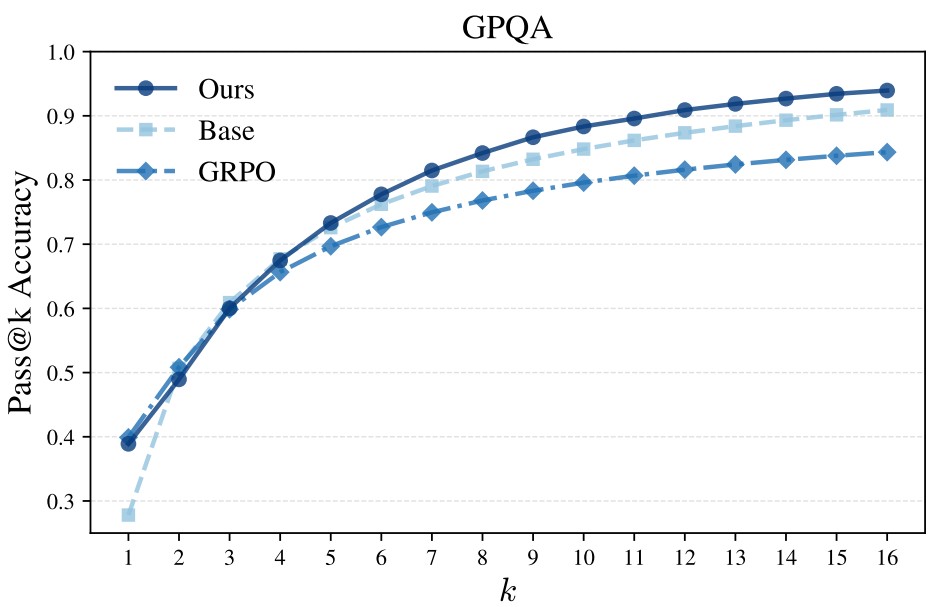

Figure 9: **Pass@$k$ performance on GPQA (Qwen2.5-Math-7B)**.

*Can you solve the following math problem? Write $\frac{3}{20}$ as a decimal. Please reason step by step, and put your final answer within $\backslash boxed\{\{\}\}$. Correct answer:* $\boxed{0.15}$

| System | Answer |
|---|---|
| Ours | To convert the fraction $\frac{3}{20}$ to a decimal, we can follow these steps: 1. Identify the numerator and the denominator of the fraction. In this case, the numerator is 3 and the denominator is 20. 2. Divide the numerator by the denominator. In this case, we divide 3 by 20. 3. Perform the division. The result is 0.15. 

 Therefore, the fraction $\frac{3}{20}$ as a decimal is $\boxed{0.15}$. |
| GRPO | To convert the fraction $\frac{3}{20}$ to a decimal, we can follow these steps: 
 1. **Understand the fraction**: The fraction $\frac{3}{20}$ means 3 divided by 20. 
 2. **Perform the division**: We need to divide 3 by 20. Since 3 is less than 20, we will add a decimal point and zeros to 3 to make the division possible. 
 3. **Set up the division**: Write 3 as 3.000 (adding three zeros for precision). 
 4. **Divide**: - 20 goes into 30 once (since $20 \times 1 = 20$). - Subtract 20 from 30 to get a remainder of 10. - Bring down the next 0 to make it 100. - 20 goes into 100 five times (since $20 \times 5 = 100$). - Subtract 100 from 100 to get a remainder of 0. 
 5. **Combine the results**: The quotient from the division is 0.15. 

 Therefore, the decimal representation of $\frac{3}{20}$ is $\boxed{0.15}$. |

Table 3: Comparison on MATH500: Qwen2.5-Math-7B. We present an example where our method and GRPO are correct.

*The Fib4 number sequence is a sequence similar to the Fibonacci sequence defined as: fib4(0)=0, fib4(1)=0, fib4(2)=2, fib4(3)=0, and for $n \geq 4$, fib4(n) = fib4(n-1) + fib4(n-2) + fib4(n-3) + fib4(n-4). Write a function to efficiently compute the n-th element without recursion.*

| System | Completion | Passed |
|---|---|---|
| Ours | ```python
# Base cases
if n == 0:
    return 0
elif n == 1:
    return 0
elif n == 2:
    return 2
elif n == 3:
    return 0

# Initialize the first four elements
fib4_values = [0, 0, 2, 0]

# Compute the rest of the sequence iteratively
for i in range(4, n+1):
    next_value = (fib4_values[i-1] + fib4_values[i
     -2] +
                  fib4_values[i-3] + fib4_values[i
     -4])
    fib4_values.append(next_value)

# Return the n-th element
return fib4_values[n]
``` | true |
| GRPO | ```python
a, b, c, d = 0, 0, 2, 0
for _ in range(n):
    a, b, c, d = b, c, d, a + b + c + d
return d
``` | false |

Table 4: HumanEval comparison on Phi-3.5-mini-instruct.

*Can you solve the following math problem? What integer $n$ satisfies $0 \leq n < 18$ and*
$$n \equiv -11213141 \pmod{18} \text{ ?}$$

*Please reason step by step, and put your final answer within $\backslash boxed\{\{\}\}$.* Correct answer: $\boxed{13}$

| Method | Answer |
|---|---|
| Ours | First, we perform the division of 11213141 by 18:

$$11213141 \div 18 \approx 622952.277777\ldots$$

The integer part of the quotient is 622952. Now, we multiply 622952 by 18 to find the product:

$$622952 \times 18 = 11213136$$

Next, we subtract this product from 11213141 to find the remainder:

$$11213141 - 11213136 = 5$$

So, the remainder when 11213141 is divided by 18 is 5. This means:

$$11213141 \equiv 5 \pmod{18}$$

Therefore:

$$-11213141 \equiv -5 \pmod{18}$$

Since we need a non-negative remainder, we convert $-5$ to a positive equivalent by adding 18:

$$-5 + 18 = 13$$

Thus:

$$-11213141 \equiv 13 \pmod{18}$$

The integer $n$ that satisfies $0 \leq n < 18$ and $n \equiv -11213141 \pmod{18}$ is: $\boxed{13}$. |
| GRPO | Step 1: Find the remainder when 11213141 is divided by 18. We can use the property of modular arithmetic that states $a \equiv b \pmod{m}$ if and only if $a \bmod m = b \bmod m$. So, we need to compute $11213141 \bmod 18$.
To simplify this, we can use the fact that the remainder of a number when divided by 18 is the same as the remainder of the sum of its digits when divided by 18 (although this is not always the most efficient method, it can be used here for simplicity).
First, let's sum the digits of 11213141:

$$1 + 1 + 2 + 1 + 3 + 1 + 4 + 1 = 14.$$

Now, we need to find the remainder when 14 is divided by 18. Since 14 is less than 18, the remainder is simply 14. Therefore,

$$11213141 \equiv 14 \pmod{18}.$$

Step 2: Find the remainder when $-11213141$ is divided by 18. Since $11213141 \equiv 14 \pmod{18}$, it follows that

$$-11213141 \equiv -14 \pmod{18}.$$

We need to convert $-14$ to a positive remainder between 0 and 17. To do this, we add 18 to $-14$:

$$-14 + 18 = 4.$$

Therefore,

$$-11213141 \equiv 4 \pmod{18}.$$

The integer $n$ that satisfies $0 \leq n < 18$ and $n \equiv -11213141 \pmod{18}$ is $\boxed{4}$. |

Table 5: MATH500 comparison between our sampling algorithm and GRPO for Qwen2.5-Math-7B. Here is an example where GRPO gets an incorrect answer, while our sampling algorithm succeeds. Our sample answer uses a distinct method altogether.

