# OpenReview forum: "Reasoning with Sampling: Your Base Model is Smarter Than You Think"
_ICLR.cc/2026/Conference — ICLR 2026 Oral_

### Official Review · Reviewer_nAW4 · 2025-10-17

**Soundness:** 4
**Presentation:** 3
**Contribution:** 3
**Rating:** 8
**Confidence:** 4

**Summary:**

a new inference scaling method that can match the RL performance by spending cost during decoding, but does not require training, verifiers, datasets.

**Strengths:**

- the method is conceptually simple and straightforward
- the method does not require any training, data or verifier, and is entirely formed by the base model's own distribution

**Weaknesses:**

- need to pay cost for each sample, and if not paying for any inference scaling cost degenerates to sampling from p.
- the evaluated benchmarks are rather older. do we expect the method to be still tractable with harder datasets that require longer CoT lengths? at least evaluate AIME25 or better HMMT25/BRUMO25, which are common math reasoning benchmarks to be tested on

**Questions:**

- why was there a decrease in performance when increasing the MCMC steps, was this more noisy as high likelihood samples could also be wrong? could be good to provide distributional stats for plot 5 b.
- did you try your method on top of GRPO'd models, what will happen?

---

> ### Author Response · Authors · 2025-11-21
>
> We thank the reviewer for their insightful questions!
>
> **"Why was there a decrease in performance when increasing the MCMC steps... ?"**
>
> The power sampling skew towards high likelihood sequences in general pushes accuracy up, but can be subject to inaccuracies in the single-shot case which appears as the fluctuations in the MCMC scaling graph. In general, the accuracy fluctuates with a variance around ~2% in accuracy.
>
> Just to provide additional distributional information about power sampling (separate from the scaling graph), in the updated paper in Section 5.3 and Appendix A.4, we plot pass@k performance from k=1 to 16 (i.e. over k samples, is at least one answer correct). Our pass@k strictly supersedes the base model as well as GRPO starting from reasonably small k, so in general, the sampling distribution is stronger in reasoning across multiple samples, not just one sample. We include a plot here: https://ibb.co/dwpCpfTK.
>
>
> **"Did you try your method on top of GRPO'd models, what will happen?"**
>
> Great question! We ran experiments testing power sampling on GRPO models. We found that in-domain (on MATH500), we could experience a slight boost from +0.5% to +2% accuracy. In particular, on MATH500, Qwen2.5-Math jumps from 78.5% to **79%** accuracy, while Qwen2.5-base jumps from 74% to **76%**. On out-of-domain tasks, performance remains the same. We have included these results in the appendix.

---

### Official Review · Reviewer_4b7A · 2025-10-26

**Soundness:** 4
**Presentation:** 4
**Contribution:** 4
**Rating:** 8
**Confidence:** 3

**Summary:**

This work investigates how distribution sharpening can improve reasoning without additional training. The authors introduce a simple iterative sampling algorithm that leverages the base model’s own likelihoods to refine generation. Across three different base models and a range of benchmarks, the proposed method achieves competitive performance with alignment-trained models—while requiring no post-training or additional data.

**Strengths:**

- The paper reframes alignment and reasoning improvement as a distribution-sharpening problem, bridging the gap between post-training RL methods and pure inference-time algorithms.
- The proposed iterative sampling algorithms is simple, straightforward but exceptionally effective.
- The authors conducted solid experiences across 3 base models and several benchmarks to demonstrate the proposed method's effectiveness.

**Weaknesses:**

- The experiments are only conducted on small models such as Qwen2.5-7B. Experiments on more advanced models and larger models can further solidify the empirical exploration. But it is understandable given the resources in academia.
- The evaluation spans four benchmarks, which, while representative, do not fully capture the diversity of reasoning and generation tasks.

**Questions:**

- Why does applying GRPO to Phi-3.5-mini-instruct result in performance degradation on two math benchmarks? The authors should further analysis the underlying cause of this phenomenon.

---

> ### Author Response · Authors · 2025-11-21
>
> We thank the reviewer for their positive assessment and great questions!
>
> **"Why does applying GRPO to Phi-3.5-mini-instruct result in performance degradation on two math benchmarks?"**
>
> If you’re referring to the drop in accuracy in HumanEval (the other benchmarks all go up with GRPO), a good number of errors could be attributed to issues in indent formatting (although fundamentally incorrect algorithms are also generated as in Table 2). Around 75% of failures are at least due to formatting issues (which can also have algorithmic issues), while 25% of failures are attributed to algorithmic failures (like Table 2) (see also: https://ibb.co/nMrg9Nry). This sustains a pattern from the base model, where around 80% of failures in standard sampling are due to this unindent issue. These errors do not show up in the MCMC generated responses.
>
> We include some commentary about this in Appendix A.4.

---

### Official Review · Reviewer_mX2n · 2025-10-31

**Soundness:** 4
**Presentation:** 4
**Contribution:** 4
**Rating:** 6
**Confidence:** 4

**Summary:**

This paper proposes a training-free method to improve reasoning in large language models (LLMs) by sampling from a sharpened distribution of the base model. Inspired by ideas from Markov chain Monte Carlo (MCMC), the authors introduce an inference-time sampling algorithm that approximates sampling from a power distribution, where higher-likelihood sequences under the base model are exponentially upweighted.

**Strengths:**

- The paper offers a fresh conceptual lens by interpreting RLVR-based improvements as a form of distribution sharpening, reframing how we think about reasoning emergence in LLMs.
- The authors demonstrate that a simple, training-free algorithm can match or surpass GRPO-trained models across multiple benchmarks and model families.

**Weaknesses:**

As noted by the authors, the method can only exploit existing capabilities of the base model—it cannot exceed its representational capacity. A deeper discussion or analysis of the top-k ceiling would help contextualize the results.

**Questions:**

- Have the authors evaluated Power Sampling on open-ended tasks like story writing or dialogue? If not, what properties of reasoning tasks (e.g., verifiability, objective correctness) make sharpening effective?
- Is a higher base model likelihood always desirable? Could overly sharp distributions lead to degenerate or repetitive text, as seen in low-temperature sampling?
- In my own experiments, sharpened sampling does not seem to improve autoregressive image generation. Do the authors believe that *reasoning tasks* uniquely satisfy conditions that make $p^{\alpha}$ beneficial?

---

> ### Author Response · Authors · 2025-11-21
> **Part I: Top-k and High Likelihoods**
>
> We sincerely thank the reviewer for their immensely thoughtful questions and comments! We address them below:
>
> **"A deeper discussion or analysis of the top-k ceiling would help contextualize the results."**
>
> We have updated the paper with results examining the pass@k performance (success if 1 of k samples is correct) of our sampler on Qwen2.5-Math-7B.
>
> Broadly, we find that our pass@k performance does not deteriorate as with GRPO; we are able to achieve strong single-shot reasoning performance without losing diversity over multiple samples, maintaining strong multi-shot reasoning as well.
> Power sampling produces a pass@k curve that is strictly stronger than that of both the base model and GRPO across our benchmarks (MATH500, HumanEval, GPQA). We include these results in Section 5.3 of the updated paper as well as Appendix A.4. We also provide the plot for MATH500 here: https://ibb.co/dwpCpfTK.
>
> **"Have the authors evaluated Power Sampling on open-ended tasks... What properties of reasoning tasks (e.g., verifiability, objective correctness) make sharpening effective?"**
>
> AlpacaEval 2.0 is one open-ended task we evaluated on (with responses being judged for generic helpfulness), where we saw power sampling could also improve upon GRPO (as well as the base model). In general, calibration of the pretrained model with respect to the downstream eval makes sharpening the most effective (i.e. confidence/negative entropy and likelihoods being correlated with either more correctness or higher quality). Natural verifiability such as correctness is likely propagated in pretraining towards higher likelihoods if the base model tends to be trained on correct traces, which is one likely reason behind the effectiveness of sharpening for reasoning.
>
> **"Is a higher base model likelihood always desirable? Could overly sharp distributions lead to degenerate or repetitive text, as seen in low-temperature sampling?"**
>
> You are correct to point out that the ultimate target is not higher base model likelihoods, but correct reasoning traces. For example, very difficult reasoning problems might have solutions that lie in low-likelihood regions under the base model: novel sampling techniques beyond the current RL paradigm and likelihood-sharpening with power sampling are needed to access these gains in reasoning.
>
> We did not observe excessively repetitive text during generation for reasoning tasks; however, the emergence of repetitive text may occur in more general settings. We use low-temperature sampling as a proposal distribution, so without sufficient mixing, repetition from the proposal distribution can in principle propagate to the power sampling output. However, adding a penalty against repetition during the proposal generation (as in [1]) can help mitigate this issue.
>
>
> --------------------------------------------------------
> **References**
>
> [1]: Ginart et. al. 2025. "LZ Penalty: An information-theoretic repetition penalty for autoregressive language models"

---

> ### Author Response · Authors · 2025-11-21
> **Part II: Autoregressive Image Generation**
>
> **"In my own experiments, sharpened sampling does not seem to improve autoregressive image generation. Do the authors believe that reasoning tasks uniquely satisfy conditions that make beneficial?"**
>
> We experimented with power sampling for autoregressive image generation at a small scale. We looked at “classifier-free guidance” (CFG), where the attempt is to do sharpened sampling by targeting sampling from the distribution $p(x|c)^{\gamma} \cdot p(x)^{1 - \gamma}$, where $\gamma > 1$ and $c$ is a class-condition label. Currently, classifier-free guidance samplers are analogous to classic “low-temperature sampling” for text, where the logits are given by $\gamma \cdot $ log$(x_{T+1} | c, x_1 \dots x_T) + (1-\gamma) \cdot $ log$(x_{T+1} | x_1 \dots x_T)$ for each next token prediction $x_{T+1}$. As we mention in our paper, this does not sample from the true sharpened distribution, but nonetheless, has been empirically shown to improve image quality (see [2]).
>
> In other words, sharpening can help autoregressive image generation in this formulation. If the question is whether this “per-token” approximation vs. true target distribution sampling with MCMC has any impact on image quality, at a small scale we found quality hard to distinguish. We generated 100 images on a random subset of ImageNet class labels and computed the FID/CMMD scores generated by the standard process, CFG, and our MCMC sampling targeting the true CFG distribution. While CFG and MCMC outperform standard generation, more comprehensive analysis is needed to see if there is a noticeable difference between the two.
>
> | Method | FID ↓    | CMMD ↓  |
> |--------|----------|---------|
> | **STD**  | 185.669 | 1.348   |
> | **CFG**  | 170.547 | 0.781   |
> | **MCMC** | 174.198 | 0.937   |
>
> For reasoning, there is a large difference between the “per-token” approximation vs. power sampling. As we mention in the paper, one reason is due to the empirical phenomenon of pivotal tokens (discovered concurrently by Phi-4 team [3] and [4]), which demonstrates that certain singular tokens during math/reasoning tasks heavily influence the final correctness of the response. Certain tokens can trap the generation into an incorrect response; power sampling helps mitigate this as it avoids tokens that have high “average weight” but with only (several) low-likelihood futures in preference of tokens that have high-likelihood completions. For images, token generation is perhaps less sequentially correlated, which might diminish the difference between MCMC sampling and next-token distribution approximations for autoregressive sampling.
>
>
> ----------------------------------------------
> **References**
>
> [2]: Yu et. al. 2024. "Randomized Autoregressive Visual Generation"
>
> [3]: Abdin et. al. 2024. "Phi-4 Technical Report"
>
> [4]: Li et. al. 2025. "Blink of an eye: a simple theory for feature localization in generative models"

---

### Official Review · Reviewer_L8G7 · 2025-11-01

**Soundness:** 4
**Presentation:** 3
**Contribution:** 4
**Rating:** 8
**Confidence:** 4

**Summary:**

- The paper shows that strong reasoning performance can be achieved from base LLMs without any additional training by changing how we sample from them.
- It introduces a power sampling algorithm that draws from a sharpened distribution $p_{\alpha}(\mathbf{x}) \propto p(\mathbf{x})^\alpha$, emphasizing higher-likelihood reasoning traces.
- The method uses an MCMC-style resampling process to approximate sampling from this distribution efficiently at inference time.
- Across several benchmarks such as MATH500, HumanEval, and GPQA, this training-free approach matches or surpasses GRPO-based post-training.
- The results suggest that base models already encode strong reasoning abilities, and that better inference-time sampling can unlock them without retraining.

**Strengths:**

- The method achieves RL-level reasoning performance without any additional training, data, or reward signals.
- It provides a clear theoretical analysis showing why power sampling differs fundamentally from conventional low-temperature sampling.
- The algorithm is simple, mathematically principled, and broadly applicable to existing base language models.
- The work reframes reasoning enhancement as an inference-time sampling problem rather than a post-training or reward-learning problem.

**Weaknesses:**

- It depends on tuning the power factor, block size, and number of MCMC steps, which may limit plug-and-play usability.
- The algorithm provides no theoretical guarantee of convergence to the true power distribution in large sequence spaces.
- The paper evaluates only single-shot responses and does not test multi-turn reasoning or chain-of-thought extensions.

**Questions:**

- How can $\alpha$ be selected practically and automatically without extensive tuning?
- Does power sampling remain effective for longer reasoning chains or multi-turn dialogues?
- How does the diversity of outputs change with increasing $\alpha$ and MCMC steps, and is there a risk of collapsing to repetitive or overconfident generations?
- Have you considered evaluating the proposed method on more challenging or diverse reasoning benchmarks such as MMLU?
- Could you provide insights into why $\alpha = 4.0$ consistently works well across tasks?
- Have you considered using tempered transitions within your MCMC framework to improve chain mixing, especially when sampling from highly sharpened distributions for long reasoning sequences?
- In Line 17, Line 46, and more, capabilites -> capabilities
- In Line 356, algorihtms -> algorithms
- The figures use multiple shades of blue that differ only in brightness, making it difficult to distinguish between lines or bars, especially when printed or viewed in grayscale. The authors are encouraged to adopt a more distinguishable color palette.
- Use \citep and \citet appropriately.
- Ensure that the citation formats are consistent, the capitalization is correct, and the reference information is up-to-date.

**Details Of Ethics Concerns:**

There is no particular ethical concern.

---

> ### Author Response · Authors · 2025-11-21
>
> We thank the reviewer for their helpful and insightful comments!
>
> **"How can $\alpha$ be selected practically and automatically without extensive tuning?"**
>
> We did not extensively tune the $\alpha$ parameter for our experiments; we tried $\alpha = 2$ and $\alpha = 4$ and found $\alpha = 4$ works quite well out-of-the-box for the reasoning tasks we evaluated on. In general, from Figure 5, as long as $\alpha > 2$, we can achieve reasonable performance up to $\pm 2$-$3$% in accuracy, so in practice this parameter is robust to imperfect tuning.
>
>
> **"Does power sampling remain effective for longer reasoning chains or multi-turn dialogues?"**
>
> The GPQA Diamond reasoning chains are relatively long compared to the MATH500 benchmarks; while MATH500 has around an average length of 670 tokens, GPQA has a length of 900 tokens, and we see similar boosts from MCMC sampling. Because the sampling is done blockwise, this makes our MCMC sampling less prone to falling short on longer sequences compared to simply running Metropolis-Hastings on the full sequence length directly.
>
> Multi-turn dialogues are an interesting non-verifiable benchmark that we can add to the final version of this paper!
>
> **"How does the diversity of outputs change with increasing $\alpha$ and MCMC steps, and is there a risk of collapsing to repetitive or overconfident generations?"**
>
> We have updated the paper with broader discussion and experiments related to diversity in Section 5.3 and Appendix A.4. We find that with MCMC sampling, there is no collapse in diversity for $\alpha = 4$, as evidenced by our pass@k (multi-shot reasoning) performance curves superseding the base model. We have an associated plot here as well: https://ibb.co/dwpCpfTK
>
> Of course, at $\alpha = \infty$, since this is just max-likelihood sequence sampling, there is a loss of diversity, and diversity should remain the same for higher MCMC steps across $\alpha$ given the accuracy stabilizes.
>
> **"Have you considered evaluating the proposed method on more challenging or diverse reasoning benchmarks such as MMLU?"**
>
> We chose GPQA Diamond as the challenging “general knowledge” reasoning task as opposed to MMLU by looking at base model performance (Qwen achieves around 70% accuracy on MMLU as opposed to 20% accuracy on GPQA). Nonetheless, we can update MMLU as part of a broader evaluation suite in the final version of the paper!
>
>
> **"Could you provide insights into why $\alpha = 4.0$ consistently works well across tasks?"**
>
> From Figure 5, in general $\alpha > 2$ works relatively well up to $2$-$3$% changes in error. Too high of an $\alpha$ over-optimizes high-likelihood sequences, which can come at a cost in accuracy. Too low of an $\alpha$ simply resamples from the base model. $\alpha = 4$ is a middle ground that does not sharpen too strongly, maintaining a strong reasoning sampling distribution: again, see pass@k figure here (https://ibb.co/dwpCpfTK) as well as in the updated paper in Section 5.3 and Appendix A.4.
>
>
> **"Have you considered using tempered transitions within your MCMC framework to improve chain mixing, especially when sampling from highly sharpened distributions for long reasoning sequences?"**
>
> We had not implemented this, but we thank the reviewer for suggesting tempered transitions! This offers an exciting algorithmic direction to try for long-generation tasks.
>
> **Formatting/typos/citations**
>
> Thank you for pointing these out! We have fixed these in the updated draft.

---

> > ### Comment · Reviewer_L8G7 · 2025-11-25
> >
> > Thank you for the authors' clarification.
> >
> > I would like to maintain my score as Accept.

---

### Public Comment · ~Kartikeya_Bhardwaj1 · 2025-11-21
**Great work!**

Dear authors,

This is an excellent work. Novel techniques to perform reasoning without any training are much needed, particularly since RL techniques can be expensive. I wanted to bring a concurrent work to your notice: Video Reasoning without Training (https://arxiv.org/abs/2510.17045). We also focus on the similar "Reasoning without Training" idea but from a video reasoning perspective. Our technique (called V-Reason) is based on an inference time optimization, so it is orthogonal to sampling based methods. We also show that inference time for our method is significantly faster than the RL-trained models simply by reducing output tokens.

We plan to discuss your paper in our work as a concurrent method. I would greatly appreciate if you could discuss our paper in a similar vein. The community needs to pay attention to this novel topic.

All the best,
Kartikeya

---

### Author Response · Authors · 2025-12-02
**Summary of Responses for the AC's Reference**

We thank the reviewers for their engagement and thoughtfulness during the review process. We appreciate our reviewers commendations on our methods' "simplicity" (**naW4**, **mX2n**, **L8G7**) but "exceptional effectiveness" (**4b7a**).

We summarize our responses to the reviewers’ main points and questions below:

**Beyond Single-Shot Reasoning: Pass@k and Diversity**

In the prior version of the paper, we show that our sampling technique leads to large boosts in single-shot reasoning, comparable to RL. As per Reviewer **mX2n**'s, Reviewer **L8G7**'s, and Reviewer **nAW4**'s suggestions, we now include experiments that demonstrate that our sampling technique does not lead to collapse in diversity and in fact demonstrates stronger pass@k performance/multi-shot reasoning than both GRPO and the base model itself (see: https://ibb.co/dwpCpfTK and Section 5.3; Appendix A.4). In particular, we show across our domains (MATH500, HumanEval, GPQA), this pass@k accuracy curve up until k=16 (where convergence with the base model occurs)

**Power Sampling on GRPO'd Models**

In response to Reviewer **nAW4**'s question, we ran experiments testing power sampling on GRPO models. We found that in-domain (on MATH500), we could experience a slight boost from **+0.5%** to **+2%** accuracy. In particular, on MATH500, Qwen2.5-Math jumps from **78.5%** to **79%** accuracy, while Qwen2.5-base jumps from **74%** to **76%**. On out-of-domain tasks, performance remains the same. We have included these results in the appendix.

**Selecting Hyperparameters**

We address hyperparameter selection questions from Reviewer **L8G7**. We show in the paper that $\alpha$ parameter is robust to un-careful tuning as long as it is greater than $2$, at least for single-shot reasoning performance. Without careful tuning, we found $\alpha = 4$ works very well out-of-the-box, and has strong distributional properties especially with a better pass@k performance. Too high of an $\alpha$ over-optimizes high-likelihood sequences, which can come at a cost in accuracy, while too low simply resamples from the base model. $\alpha = 4$ is a middle ground that does not sharpen too strongly, maintaining a sampling distribution.

**Why Sharpening Can Work Well for Reasoning and Extensions to Other Domains**

Reviewer **mX2n** inquired about why sharpening should be expected to work well for reasoning and whether it could be extended to autoregressive image generation. For reasoning, there is a large difference between the “per-token” approximation vs. power sampling, but both are “sharpening” the base distribution. As we mention in the paper, one reason for this difference is due to the empirical phenomenon of pivotal tokens, which are tokens that strongly influence the correctness of the reasoning trace. Power sampling avoids pivotal tokens that traps outputs in low-likelihood completions, while low-temperature sampling does not.

We also run some experiments in the image domain, looking at autoregressive image generation. We demonstrate that sharpening can lead to higher image quality, although at the scale we tested at (100 sample images), there isn’t too much difference between sharpening via naive classifier-free guidance (“low-temperature sampling”) vs. via proper MCMC sampling. This could be due to the lower sequential dependence in image generation, but larger scale experiments are needed to say something more definitive.


We include further details in the follow-up comment for the AC's reference.

---

> ### Author Response · Authors · 2025-12-03
> **Further Experimental Details for the AC**
>
> Here we add some additional details about the experiments we ran in response to the reviewer's questions:
>
> **Pass@k and Diversity**
>
> We have updated the paper with results examining the pass@k performance (success if 1 of k samples is correct) of our sampler on Qwen2.5-Math-7B.
>
> Broadly, we find that our pass@k performance does not deteriorate as with GRPO; we are able to achieve strong single-shot reasoning performance without losing diversity over multiple samples, maintaining strong multi-shot reasoning as well.
> Power sampling produces a pass@k curve that is strictly stronger than that of both the base model and GRPO across our benchmarks (MATH500, HumanEval, GPQA). We include these results in Section 5.3 of the updated paper as well as Appendix A.4. We also provide the plot for MATH500 here: https://ibb.co/dwpCpfTK.
>
> **Sharpened Autoregressive Image Generation**
>
> We experimented with power sampling for autoregressive image generation at a small scale. We looked at “classifier-free guidance” (CFG), where the attempt is to do sharpened sampling by targeting sampling from the distribution $p(x|c)^{\gamma} \cdot p(x)^{1 - \gamma}$, where $\gamma > 1$ and $c$ is a class-condition label. Currently, classifier-free guidance samplers are analogous to classic “low-temperature sampling” for text, where the logits are given by $\gamma \cdot $ log$(x_{T+1} | c, x_1 \dots x_T) + (1-\gamma) \cdot $ log$(x_{T+1} | x_1 \dots x_T)$ for each next token prediction $x_{T+1}$. As we mention in our paper, this does not sample from the true sharpened distribution, but nonetheless, has been empirically shown to improve image quality (see [2]).
>
> In other words, sharpening can help autoregressive image generation in this formulation. If the question is whether this “per-token” approximation vs. true target distribution sampling with MCMC has any impact on image quality, at a small scale we found quality hard to distinguish. We generated 100 images on a random subset of ImageNet class labels and computed the FID/CMMD scores generated by the standard process, CFG, and our MCMC sampling targeting the true CFG distribution. While CFG and MCMC outperform standard generation, more comprehensive analysis is needed to see if there is a noticeable difference between the two.
>
> | Method | FID ↓    | CMMD ↓  |
> |--------|----------|---------|
> | **STD**  | 185.669 | 1.348   |
> | **CFG**  | 170.547 | 0.781   |
> | **MCMC** | 174.198 | 0.937   |
>
>
> **GRPO on Phi 3.5-mini and Degradation on HumanEval**
>
> This is in response to Reviewer **4b7a**'s question about GRPO resulting in a drop in accuracy in HumanEval (the other benchmarks all go up with GRPO) for Phi 3.5. A good number of errors could be attributed to issues in indent formatting (although fundamentally incorrect algorithms are also generated as in Table 2). Around 75% of failures are at least due to formatting issues (which also have algorithmic issues), while 25% of failures are attributed to algorithmic failures (like Table 2) (see also: https://ibb.co/nMrg9Nry). This sustains a pattern from the base model, where around 80% of failures in standard sampling are due to this unindent issue. These errors do not show up in the MCMC generated responses.
>
> We include some commentary about this in Appendix A.4.
>
> **Open Ended and Long Generation Tasks**
>
> Reviewers **L8G7** and **mX2n** inquired about open-ended tasks and longer reasoning chains. AlpacaEval 2.0 is one open-ended task we evaluated on (with responses being judged for generic helpfulness), where we saw power sampling could also improve upon GRPO (as well as the base model). In general, calibration of the pretrained model with respect to the downstream eval makes sharpening the most effective (i.e. confidence/negative entropy and likelihoods being correlated with either more correctness or higher quality).
>
> For long generation tasks, the GPQA Diamond reasoning chains are relatively long compared to the MATH500 benchmarks; while MATH500 has around an average length of 670 tokens, GPQA has a length of 900 tokens, and we see similar boosts from MCMC sampling. Because the sampling is done blockwise, this makes our MCMC sampling less prone to falling short on longer sequences compared to simply running Metropolis-Hastings on the full sequence length directly.
>
> **Collapsed Generations**
>
> Reviewers **L8G7** and **mX2n** inquire about the potential for repetitive generations. We did not observe excessively repetitive text during generation for reasoning tasks. However, we use low-temperature sampling as a proposal distribution, so without sufficient mixing, repetition from the proposal distribution can in principle propagate to the power sampling output. However, adding a penalty against repetition during the proposal generation (as in [1]) can help mitigate this issue.
>
> --------------------
> References
>
> [1]: Ginart et. al. 2025. "LZ Penalty: An information-theoretic repetition penalty for autoregressive language models"

---

### Meta-Review · Area_Chair_i8SQ · 2026-01-07

**Summary:**

This paper proposes a training-free inference-time method, power sampling, that substantially improves reasoning performance of base LLMs by sampling from a sharpened version of the model’s own distribution. Reviewers broadly agree that the work is conceptually novel, mathematically well-motivated, and empirically strong. Across multiple reasoning benchmarks, the method matches or exceeds the performance of GRPO-style post-training while requiring no additional data, rewards, or training. The main concerns raised relate to hyperparameter tuning, scalability to longer or more diverse tasks, theoretical guarantees of convergence, and the scope of empirical evaluation. Overall, the reviews converge on the view that this is a strong and impactful contribution that reframes reasoning improvement as an inference-time problem.

**Reviewer Concerns:**

**Concerns largely addressed by the rebuttal:**

Hyperparameter sensitivity and robustness: The authors clarified that the power parameter is robust across a reasonable range and requires minimal tuning in practice, supported by new empirical analysis.

Diversity and degeneration risks: Additional pass@k experiments and discussion convincingly show that power sampling does not collapse diversity at practical settings and often improves multi-sample performance relative to both the base model and GRPO.

Effectiveness on longer reasoning chains: Evidence from GPQA (with longer average sequences) and blockwise MCMC sampling mitigates concerns about degradation on longer generations.

Top-k ceiling and diversity trade-offs: The added pass@k analysis helps contextualize how the method balances sharpening and diversity better than RL-based approaches.

Formatting, typos, and presentation issues: These were acknowledged and corrected.

**Concerns partially or not fully addressed:**

Scalability to larger models: Experiments are limited to relatively small model sizes, which leaves open questions about behavior at larger scales.

Inference cost: The additional decoding-time cost is inherent to the method and remains a trade-off rather than a resolved issue.

**Reviewer Scores:**

Reviewer L8G7: Remains at 8 (Accept); explicitly confirmed no score change.

Reviewer mX2n: Likely remains around 6 or increase it to 8, given the added analyses and clarifications.

Reviewer 4b7A: Likely remains at 8 (Accept); concerns were minor and largely contextual.

Reviewer nAW4: Likely remains at 8 (Accept); rebuttal addressed performance fluctuations and extensions.

---

### Decision · Program_Chairs · 2026-01-26

Accept (Oral)